# Salicylic Acid Signals Plant Defence against Cadmium Toxicity

**DOI:** 10.3390/ijms20122960

**Published:** 2019-06-18

**Authors:** Bin Guo, Chen Liu, Yongchao Liang, Ningyu Li, Qinglin Fu

**Affiliations:** 1Institute of Environment, Resource, Soil and Fertilizer, Zhejiang Academy of Agricultural Sciences, Hangzhou 310021, China; liuchen@mail.zaas.ac.cn (C.L.); liningyu259@126.com (N.L.); 2Key Laboratory of Environment Remediation and Ecological Health, Ministry of Education, Hangzhou 310058, China; ycliang@zju.edu.cn; 3College of Environmental and Resource Sciences, Zhejiang University, Hangzhou 310058, China

**Keywords:** salicylic acid, cadmium, reactive oxygen species, antioxidant defence system, glutathione

## Abstract

Salicylic acid (SA), as an enigmatic signalling molecule in plants, has been intensively studied to elucidate its role in defence against biotic and abiotic stresses. This review focuses on recent research on the role of the SA signalling pathway in regulating cadmium (Cd) tolerance in plants under various SA exposure methods, including pre-soaking, hydroponic exposure, and spraying. Pretreatment with appropriate levels of SA showed a mitigating effect on Cd damage, whereas an excessive dose of exogenous SA aggravated the toxic effects of Cd. SA signalling mechanisms are mainly associated with modification of reactive oxygen species (ROS) levels in plant tissues. Then, ROS, as second messengers, regulate a series of physiological and genetic adaptive responses, including remodelling cell wall construction, balancing the uptake of Cd and other ions, refining the antioxidant defence system, and regulating photosynthesis, glutathione synthesis and senescence. These findings together elucidate the expanding role of SA in phytotoxicology.

## 1. Introduction

Cadmium (Cd) is one of the most toxic pollutants for all living organisms with a long biological half-life [1,2]. It exists in soil naturally by weathering of the parent rocks and accumulates through anthropogenic activities, such as deposition of airborne Cd from smelting and mining, wastewater irrigation, application of Cd-contaminated phosphate fertilizers and soil amendments [3,4,5,6]. Cd is a non-essential element for plants, and its uptake by plants has posed a serious health issue to humans through the food chain. For this reason, Cd phytotoxicity is a major subject in current research on plant biology [7]. 

Cadmium can be absorbed easily by plant roots, and transported to plant shoots, leading to various visible toxic symptoms, such as growth retardation, wilting, leaf chlorosis, and cell death [8,9]. The mechanisms of Cd toxicity include replacing and inactivating essential elements and destroying protein structure, thereby interfering with various physiological processes, such as photosynthesis [10], respiration [11], element assimilation [12], and cell division [13]. To defend against Cd injury, plants utilize many coordinated strategies, such as binding Cd to the cell wall [14], vacuolar retention through chelation via phytochelatins (PCs) [15] and upregulation of the antioxidant system [16]. Since all of the above mechanisms are closely cross-linked, it is important for plant survival that these metabolic reactions are adjusted accordingly via regulating signals.

Salicylic acid (SA), a simple phenolic molecule, has long been recognized as a potent phytohormone that regulates plant development and defence in higher plants [17]. The synthesis of SA occurs by two distinct and compartmentalized pathways. One pathway derives from phenylalanine and takes place in the cytoplasm. First, phenylalanine ammonia-lyase (PAL) converts phenylalanine into cinnamic acid; cinnamic acid is then decarboxylated to form benzoic acid and finally undergoes 2-hydroxylation to generate SA [18]. Mutation of PAL genes in *Arabidopsis* results in 50% decrease in pathogen-induced SA accumulation, suggesting that the PAL pathway indeed contributes to SA biosynthesis [19]. Another biosynthetic pathway is through isochorismate synthase that catalyses the conversion of chorismate into isochorismate. [20]. In this pathway, SA is generated in chloroplasts from chorismate by the synthesis of two isochorismate synthases, *ICS1* and *ICS2* [21]. Analysis of SA-deficient mutants, *sid2*, revealed that loss of ICS1 suppresses the pathogen-induced SA accumulation [20], whereas loss of both ICSes results in further reduction of SA concentration [21].

It has been well established that SA is a key signal regulating local and systemic plant defence responses against pathogens [20,22]. In this signalling pathway, SA-binding proteins, such as SABP2, NPR3, NPR4 and NPR1 with high affinity for SA, are considered to be SA receptors that induce the expression of pathogenesis-related (PR) proteins and trigger systemic acquired resistance (SAR) [23]. Moreover, SA has shown important roles in mediating plant responses to abiotic stresses [24], including drought [25], chilling [26], osmotic stress [27], thermogenesis [28] and heavy metal toxicity [29]. Recently, the beneficial roles of SA in enhancing plant Cd tolerance, which has been reported in a wide range of plant species, have drawn much attention. However, the negative effect of SA was also noted in castor bean seedlings in which pretreatment with SA aggravated Cd damage [30]. The effects of SA on Cd-stressed plants depend on many aspects, including the application mode, the concentrations of Cd and SA and endogenous SA in the tested plants as well as the different species and developmental stages of the plants. This review covered recent studies on (1) modes of SA treatment, (2) the diverse physiological roles of SA in reducing Cd toxicity, and (3) future prospects for research concerning SA in Cd-stressed plants.

## 2. Salicylic Acid (SA) Treatment Methods

Table 1 summarizes the recent findings in the literature on the roles of SA in Cd-stressed plants. Except for one study with castor bean and three experiments with SA-deficient mutants, most of the studies show that SA alleviates the effects of Cd damage. However, very few of these literatures are actually relevant for coming to a solution for Cd toxicity. Exogenous applications of SA are mainly performed via three modes: spraying SA onto leaves, presoaking seeds with SA solution, and hydroponic treatment of roots to SA. There were 46 international publications involved in 100 pairs of exogenous SA and Cd treatments and 5 publications involved in endogenous SA treatments. Figure 1 represented SA exposure concentrations plotted against Cd exposure concentrations. Among these studies, one in which Cd and SA treatments with 560 mg L^−1^ and 3000 µM, respectively, was excluded because the concentrations used were too high and may be misleading. Taken together, the following information can be inferred: (1) Concentrations used in Cd treatments range from 0.56 to 300 mg L^−1^ (mg kg^−1^) with a mean of 31.6 mg L^−1^ (mg kg^−1^) and a variable coefficient of 160%; (2) concentrations of SA spray treatments range from 10 to 2170 µM with a mean of 585 µM and a variable coefficient of 104%; (3) concentrations of SA used in seed presoaking treatments range from 100 to 1000 µM with a mean of 464 µM and a variable coefficient of 70%; and (4) concentrations of SA used in hydroponic exposure treatments range from 1 to 500 µM with a mean of 148 µM and a variable coefficient of 117%.

From these studies, it was determined that the levels of SA in spray or pre-soaking treatments are generally higher than those in hydroponic treatments, suggesting that the regulation of SA levels in plants through root uptake is more efficient than that in the other two modes of application. Furthermore, each mode has high variable coefficients of concentrations either for Cd or for SA treatments. It is interesting to note that partial application of SA to different organs (seeds, leaves or roots) helps plants to establish a systemic defence against Cd toxicity.

### 2.1. SA Spray

Seven studies reported that SA spray alleviated Cd damage to plants. The species included potato, peppermint, oilseed rape, melon, soybean, radish, and Indian mustard. For instance, foliar application of 600 μM SA onto potato over 10 consecutive days significantly decreased the concentrations of reactive oxygen species (ROS) in leaves and stimulated the antioxidant enzyme mechanism and the related genes (*StSABP2*, *StSOD* and *StAPX*) under 200 μM Cd stress [31]. However, the concentrations used in SA application in the literature ranged from 10 to 2170 µM, and the frequency of applications was completely different. Such large gaps make it difficult to provide guidance for implementing SA spray treatments.

### 2.2. Presoaking of Seed with SA

There are more studies (19 articles) conducted on seed presoaking than on SA spray treatments. The presoaking periods ranged from 3 to 24 h, and the presoaking levels ranged from 100 to 1000 µM. Moreover, the tested plants were mainly food crops, i.e., rice, maize, wheat, barley, and bean, and industrial crops, i.e., bluegrass, flax, alfalfa, hemp, and castor bean. Similarly, most of the presoaking reports suggested that pretreatment with SA alleviated the subsequent Cd-induced damage to plant growth and the antioxidant system. Notably, presoaking method is more practical for agricultural and/or economic use than spray treatment.

### 2.3. Hydroponic Application

Indeed, hydroponic treatment of SA is less practicable because Cd is not a problem in hydrographic environment. However, this type of studies is needed since SA or Cd treatment can be controlled more precisely and effectively, which is beneficial to the phytotoxicology research. There were 20 articles conducted on this application method. The SA concentrations ranged from 10 to 500 µM, and the treatment period ranged from 3 to 72 h. Furthermore, the timeline of SA application was different with pretreatment, simultaneous treatment and post-treatment methods. Again, hydroponic treatment of plant roots to SA mitigated Cd toxicity systemically in plants. However, more research work is needed on the transport and signalling pathways of SA through different organs under Cd stress. 

### 2.4. SA Mutants

In the tested experiments, the exposure times of Cd and SA were relatively short, and the levels of Cd and SA were relatively high. Hence, plants received the SA signal rapidly as an instant response to the “acute” Cd stress. Furthermore, information in the literature differs in terms of plant species, treatments and concentrations. All of the above uncertain issues may lead to unpredictable results. For example, when the treatment concentration of SA exceeded the appropriate range, oxidative toxicity occurred [77]. Recently, studies (5 articles) focused on the role of endogenous SA in regulating Cd tolerance by comparing SA-deficient or SA-accumulating mutants with wild-type plants. Surprisingly, although *sid2* and *NahG* are both SA-deficient *Arabidopsis* lines, these two mutants exhibited contrasting responses. In *sid2* mutants, the mutation in the gene encoding isochorismate synthase (ICS) aggravated Cd toxicity compared to the wild-type plants [72]. In contrast, the *NahG* transgenic lines manifested higher Cd resistance than the wild-type plants [74,75].

## 3. Possible Roles of SA in Alleviating Cadmium (Cd) Toxicity

Taking a comprehensive view of SA roles in response to Cd toxicity, this review focuses on the recent advances in the physiological and molecular mechanisms of the following aspects: plant growth, Cd immobilization and distribution, element assimilation, photosynthesis, ROS and the antioxidant defence system, glutathione, and senescence.

### 3.1. Plant Growth

Cadmium exposure inhibits plant growth [78]. It also causes morphological changes in leaves and roots, such as leaf chlorosis and lignification of cell walls in root tissues [79]. The Cd-induced growth inhibition is mainly due to reduction of net photosynthetic rate [80], inactivation of enzymes involved in CO_2_ fixation [81], disturbance of element metabolism [82], and induction of lipid peroxidation [83]. 

As a multifaceted phytohormone, SA mediates physiological and biochemical processes during all plant developmental stages, including seed germination, vegetative growth, seed production, and senescence [84,85]. *Arabidopsis* mutants with constitutively high levels of SA, such as *cpr5* [86], *acd6−1* [87], *agd2* [88] and *pi4kIIIβ1β2* [89] exhibit dwarf phenotype. In contrast, the SA-depleted *Arabidopsis* NahG transgenic plants have a longer vegetative stage and higher growth rate compared with wild-type plants [18]. The biochemical events involved in the regulation of cell division and growth by SA still need to be clarified [90], which may be crosslinked with AUX, ROS, Ca^2+^ and mitogen-activated protein kinase (MAPK) pathways [84].

In the presence of Cd, exogenous treatment with SA showed a positive effect on the growth of various plant species, such as flax [47], bluegrass [41], radish [36], and rice [67,68]. Soybean seedlings treated with 6 mg kg^−1^ Cd for 72 h showed retarded growth symptoms in roots, stems and leaves [91]. SA applied simultaneously at the levels of 1 μM, 10 μM and 100 μM significantly reversed these inhibitory effects. In barley, Cd exposure reduced the dry weight of shoots and roots by approximately 35%, whereas pretreatment with SA resulted in significant recovery of all the growth parameters [54].

Exogenous treatment with SA has a dose-dependent effect on plant growth, as observed in the *Arabidopsis* mutants with unnecessary or deficient SA levels. Presoaking treatment with 10 to 500 μM SA increased the germination of Kentucky bluegrass seeds, while the germination sharply declined under 1000 to 5000 μM SA treatments [41]. The Cd-induced inhibitory effects on ryegrass growth were significantly alleviated by low SA concentrations, but no effects were found for the high SA concentration [69]. Some plants, such as hemp [50], are vulnerable to high levels of SA but still benefit from SA treatment when they suffer from Cd toxicity. In castor bean seedlings, SA treatment significantly worsened plant growth in both the presence and absence of Cd [30].

Furthermore, inconsistent conclusions were drawn in experiments with *Arabidopsis* transgenic plants or mutants. The Cd-inhibited growth in *Arabidopsis* was aggravated by unnecessary SA in *snc1* mutants and mitigated by the depletion of SA in *nahG* transgenic lines [74]. It seems that endogenous SA negatively regulates *Arabidopsis* tolerance to Cd. However, the *sid2* mutants with SA-deficient phenotype showed a Cd-sensitive phenotype that manifested as having accentuated Cd-induced growth inhibition [72].

### 3.2. Cd Immobilization in the Cell Wall

The plant cell wall, as a complex composed of sugars, proteins and phenols, is the first barrier against Cd toxicity and limits Cd translocation into the cytosol [78]. The hemicellulose and pectin in the cell wall are major components for Cd deposition due to their negative charges [92]. Findings between hyperaccumulating and non-hyperaccumulating ecotypes of *Sedm alfredii* show that roots with higher cell-wall polysaccharides and activity of pectin methylesterase are more impermeable to Cd [93]. Using energy-dispersive X-ray microanalysis, Cd binding to the cell wall was directly visualized in *Thlaspi caerulescens* [14]. In addition, long-term Cd treatments may interfere synthesis and composition of cell wall, such as inducing lignification, which in turn impact Cd sequestration in a more complicated way [94]. Whereas for SA, the signalling pathway is also involved in regulation of cell wall components. Genetic research showed that *pmr4* mutants are resistant to pathogens due to mutation of a callose synthase, while blocking the SA signaling pathway restore the susceptibility, suggesting callose or callose synthase negatively regulates the SA pathway [95]. SA application altered the lignin and hemicellulose composition of cell wall in *Brachypodium distachyon* by increase in caffeic acid, ferulic acid and p-coumaric acid content [96].

Therefore, it is hypothesized that pretreatment with SA may increase Cd accumulation in the cell wall and prevent Cd translocation into other cell organelles. Until now, only a few experiments with peanut and ryegrass have shown that SA treatment reduced Cd influx by rearrangement of the cell wall composition [60,62,69]. It has been reported that either SA pretreatment or Cd treatment alone strengthened the root cell wall in rice [97]. However, it was also found that SA treatment with Cd application failed to synergistically affect the cell wall construction or the activity of enzymes related to lignin synthesis, such as polyphenol oxidase (PPO), peroxidase (POD) and PAL. This might be because the strong toxicity of Cd maximized the process of lignification, which concealed the SA signalling role in cell wall construction.

### 3.3. Cd Uptake and Translocation

The effect of SA on the construction of the cell wall is closely related to Cd uptake and translocation, but the results are still controversial. Some studies have indicated that the treatment of the plant with SA could decrease Cd assimilation and root-to-shoot translocation. For instance, SA application substantially reduced Cd uptake and ameliorated Cd-induced growth inhibition in radish roots [36]. Pretreatment of flax with SA significantly decreased the Cd levels in different organs of the plant [47]. This was also reported in other plant species, i.e., Kentucky bluegrass [41], ryegrass [62,69], alfalfa [65], oilseed rape [33], and wheat [98]. The suppression of Cd uptake and translocation might be due to SA-induced reduction in the uptake, inhibition or activation of element translocators that dislocate Cd into vacuoles [50,91]. 

However, the role of SA signalling in preventing Cd transport between plant organs is not always physiological. SA and Cd applied simultaneously increased Cd assimilation in soybean [91]. In SA-pretreated barley, tissue Cd contents were unaltered both in vacuoplasts and mesophyll cells at the organ and the whole-plant level [54]. An interesting examination of Cd translocation was conducted using a split-root system with rice [67]. In this system, an appreciable amount of Cd was transferred from Cd-treated root parts to non-Cd-treated root parts. However, pretreatment with 10 μM SA of the whole rice roots did not restrict but promoted this transport process. As for SA mutants, in *sid2* and *NahG*, SA deficiency did not influence Cd assimilation either in shoots or in roots, indicating that SA might not mediate an avoidance mechanism in plants [72,75].

### 3.4. Element Uptake

Cadmium uptake by plants involves the competition of nutrients by using the same membrane transporters, hence interfering with ionic homeostasis [54,78]. Expression studies with the Fe transporter cloned from *Arabidopsis*, *IRT1*, facilitate the Cd influx across the root-cell plasma membrane [99]. Expression of the wheat cDNA *LCT1* in *S. cerevisiae* mediates both Ca and Cd transport into the cytosol of plant cells [100]. 

The beneficial role of SA could be attributed to its maintenance of the optimal nutrition status of plants. For instance, the Cd-induced disturbance in ion uptake, including K, Fe, Ca, Mg, Mn, and Zn uptake, was alleviated by SA treatment, in flax [47], ryegrass [69], rice [101], peanut [60], Kentucky bluegrass [41] and oilseed rape [33]. This could be explained by the alterations of plasma membrane properties by SA, increasing the activity of H^+^-ATPase [102], which facilitates the assimilation of nutrients under Cd toxicity.

The effect of SA on element uptake is especially relevant for K, which plays a key role in regulating H^+^-ATPase in the plasma membrane. As early as 1981, the inhibitory function of SA was observed for K absorption in oat roots [103]. However, under Cd stress, a positive correlation between SA and K was noted in soybean seedlings [91] and the SA mutant, *sid2* [72], and SA further interfered with the uptake of Fe and Mg in roots and shoots. The interaction between SA and Cd is also involved in the assimilation of S, the key element of sulfhydryl groups that chelate Cd ions into less vulnerable organelles in plant cells. In the absence of Cd, SA treatment increased the S content of barley roots [54]. In contrast, the S levels were much higher in the *sid2* mutant leaves than in the wild-type plants under Cd-free conditions [72].

### 3.5. Photosynthesis

The mineral nutrient stress induced by Cd results in severe alterations in photosynthesis in terms of chloroplast structure, chlorophyll concentration, and activities of carboxylating enzymes [51]. Meanwhile, SA may act as an important photosynthesis regulator by the influence of RuBisCO activity, contribution to light acclimation and redox homeostasis, and the function of the stomatal switch [84]. 

Pretreatment with SA prevented Cd-induced chlorophyll destruction in maize [51], soybean [64], oilseed rape [33] and flax [47]. Conversely, depletion of SA further lowered the chlorophyll concentrations in Cd-treated *sid2* mutants [72]. The Cd-inhibited activity of RuBisCo and carbonic anhydrase was recovered by exposure to 0.1 mM SA in peppermint [32]. Moreover, SA application increased the carotenoid synthesis of soybean seedlings, whereas it decreased the flavonoid content under Cd stress [35]. The alleviating role of SA could be due to the restored K contents in leaves, strengthening stomatal closure [72], which is synchronized with the net photosynthetic rate, transpiration, capability for CO_2_ fixation and inhibition of the activities of chlorophyll-degrading enzymes [35,50,62].

In addition, non-stomatal factors signalled by SA also play roles in the maintenance of photosynthetic capacity under Cd stress. The application of SA to Cd-treated barley leaf slices significantly slowed the decreasing trends in the photosynthetic yield of photosystem II (PSII) [54]. Spraying 0.1 mM SA onto melon leaves induced an increase in Fv/Fm in Cd-stressed plants, indicating that SA improved PSII efficiency [34]. Similarly, treatment with SA in rice prevented the unnecessary energy transference from PSII to PSI induced by Cd toxicity [55]. In this study, SA increased the cyclic electron transport around PSI in thylakoid membranes and protected the Mn-cluster of the oxygen-evolving complex from Cd damage.

The photosynthetic response to SA is both dose- and species-dependent. Pretreatment with 250 and 500 μM SA in castor bean leaves failed to affect chlorophyll levels but aggravated the negative effect on photosynthesis induced by Cd [30], which might have been associated with an increase in stomatal limitation. The reduced SA level in *nahG* plants resulted in the maintenance of photosynthetic efficiency by low photoinhibition under Cd stress [71]. The response to endogenous SA in Cd tolerance might be associated with the regulation of photosynthetic electron transport, starch degradation and PSII structures at the transcriptional level.

### 3.6. Reactive Oxygen Species (ROS) and Antioxidant Defence System

The production of ROS in plant cell is an unavoidable consequence of oxygen metabolism, especially during respiration and photosynthesis [104]. The major mode of Cd toxicity in plants is to induce ROS production and results in oxidative injuries in plants [30]. Although Cd does not participate directly in cellular redox reactions (such as Fenton reaction), it can indirectly elevate ROS accumulation in the cellular environment by the disturbance of electron transport, destroying the structure of antioxidant enzymes and interfering with antioxidant molecule synthesis. Cd indirectly modulates the activity of the plasma membrane NADPH oxidase, increasing the formation of O_2_^•−^ and H_2_O_2_, which has been found in tobacco [105], rice [106], and lupine roots [107]. Meanwhile, the enhanced demand for glutathione (GSH) for Cd chelation causes rapid loss in antioxidative defence [16]. 

It is well known that SA signals plant resistance through modulation of ROS metabolism, especially H_2_O_2_. The mode of SA action involves binding directly to CAT and ascorbate peroxidase (APX), two major H_2_O_2_-scavenging enzymes [108], inhibiting their activities in plants. This finding was also confirmed in *sid2* mutants, SA-deficient *Arabidopsis* plants, whose leaf CAT activity is higher than that in wild-type plants [72]. Using the DAB staining method, the SA-induced accumulation of H_2_O_2_ was visualized in rice leaves [66]. SA may regulate H_2_O_2_ accumulation through a self-amplifying feedback loop in *Arabidopsis*. In this process, SA acts as an electron donor of CAT that effectively slows down the peroxidation cycle, hence sharply decreasing the efficiency of H_2_O_2_ elimination [108]. Another important mechanism may occur in mitochondria. SA blocks electron flow from the substrate dehydrogenises to the ubiquinone pool and triggers H_2_O_2_ generation [109]. 

Both Cd stress and SA accumulation can elevate H_2_O_2_ production in cells. Therefore, SA treatment may aggravate the oxidative stress induced by Cd toxicity. Surprisingly, a large body of studies found that SA treatment alleviates Cd-induced oxidative stress in plants. In fact, H_2_O_2_ has a dual role in plant biology as both a toxic byproduct and a key regulator against many abiotic stresses in plants [110]. The increase in H_2_O_2_ status stimulated by SA pretreatment acts as a crucial message to “set up” the antioxidant system and then induces plants to resist subsequent Cd stress. The beneficial effects of SA were found in most of the studies in which SA treatment was performed, either by spraying, presoaking, or by hydroponic incubation, in advance of the application of Cd stress (Table 1). For example, pretreatment with 10 µM SA for 72 h initially caused H_2_O_2_ accumulation in vitro in rice roots. Correspondingly, the levels of GSH, non-protein thiols (NPT), and ascorbic acid (AsA) and the activities of CAT, superoxide dismutase (SOD) and POD were elevated compared to those of non-SA treated roots during the subsequent Cd exposure period [67,68]. The expression level of selected genes (*StSABP2*, *StSOD* and *StAPX*) was enhanced in SA-treated potato plants under Cd stress [31]. Pretreatment with SA improved the antioxidant capacity and tolerance of Cd-induced oxidative stress in rice leaves, which was similar to the effects of H_2_O_2_ pretreatment [66]. Moreover, the role of SA in mitigating Cd damage was also confirmed in many other species, such as flax [39], mustard [37], oilseed rape [33], ryegrass [62,69], melon [34], etc.

Although a number of reports on the benefits of SA signalling exist in the literature, some questions have not yet been fully answered. SA treatment alleviated Cd toxicity in barley but lowered the activities of antioxidant enzymes [54]. SA could act as an oxidant at high levels due to its capability to increase H_2_O_2_ generation and oxidation of proteins in plant leaves [77]. In contrast, it was reported that SA may act as a direct scavenger to reduce excessive H_2_O_2_ injury in the pea during Cd stress [111]. In addition, SA may directly signal the expression of defence-related enzymes, such as haem oxygenase−1 (HO−1), to alleviate Cd-triggered oxidative toxicity by reestablishing redox homeostasis [64]. Moreover, the crosstalk and response pathways between SA and other phytohormones should also be noted. SA combined with NO treatment synergistically counteracted Cd-induced oxidative damage in peanut and ryegrass [60,62]. Under Cd stress, pretreatment of wheat seedlings with SA significantly promoted the synthesis of dehydrins, the abscisic acid (ABA)-signalled proteins that can neutralize and bind unnecessary H_2_O_2_ [98].

The results of the SA mutants studies are still contradictory. *snc1* mutants with high intrinsic levels of SA possess high POD activity, which can generate a large amount of ROS and manifest the Cd-sensitive phenotype [74]. In contrast, SA deficiency in NahG transgenic lines mitigates the oxidative stress induced by Cd toxicity [74,75]. In contrast, *sid2* mutants with an SA-deficient phenotype aggravated Cd-induced oxidative damage compared with their wild-type plants.

### 3.7. Glutathione and Chelation

Glutathione (γ-Glu-Cys-Gly, GSH) is one of the most important reducing equivalents in plants, protecting plants against Cd-induced oxidative damage. Furthermore, it is also a key molecular compound or a basic component of phytochelatins (PCs) involved in Cd chelation and thereby confines Cd to less sensitive organelles, such as vacuoles [15,112].

A series of genetic reports using single mutants or transformants have shown direct evidence that endogenous SA signalling is linked to GSH biosynthesis. The catalase-deficient *Arabidopsis* mutant, *cat 2*, induces SA levels and a wide range of SA-dependent responses alongside the upregulation of GSH [113]. In contrast, attenuation of GSH levels was correlated with decreased SA contents in *cat2 atrbohF* compared with *cat2* [114]. It has been shown that the GSH concentrations were much lower in Cd-stressed *sid2* leaves than in wild-type plants [72]. Low GSH levels were also found in other SA-deficient mutants, such as *npr1−1* and *mpk4−1* [115,116]. Furthermore, the exogenous application of SA has been shown to enhance S assimilation in barley roots [54] and elevate GSH content in some plant species, such as peppermint [32], flax [45], peanut [60] and rice [68]. 

The regulatory role of SA in GSH biosynthesis may be related to serine acetyltransferase (SAT) transcription, the precursor gene that catalyses cysteine formation. Increased free SA levels, both by genetics and by exogenous application, lead to an increased specific activity of SAT and GSH in *Arabidopsis* [117]. Under Cd stress, although depletion of SA in *sid2* mutants significantly enhanced the uptake of S, a key element for GSH construction, down-regulated transcription of SAT-c and SAT-p in *sid2* blocked the process of GSH biosynthesis and resulted in lower GSH levels compared with the wild-type plants [72]. Glutathione synthetase (GSHS) is a rate-limiting enzyme that catalyses the second step of GSH synthesis in plants. The expression of the Cd-induced *LcGSHS* transcript, a GSHS gene isolated from *L. chinense*, is controlled by the endogenous SA-dependent pathway and results in greater GSH accumulation and Cd tolerance in transgenic *Arabidopsis* [73]. Furthermore, SA has also been shown to mediate the synthesis of glutathione reductase (GR1), the pivotal enzyme for regenerating and maintaining GSH in the reduced state [118]. During Cd exposure, SA deficiency significantly decreased GR1 transcription in *sid2* mutants and resulted in lower GSH levels and GR activity compared with the wild-type plants [72]. However, another SA-deficient transgenic line, *NahG*, manifested high GSH accumulation and a high GSH/GSSG ratio in the presence of Cd [74,75]. These inconsistent results suggest that the role of SA in regulating GSH synthesis under Cd stress requires further investigation.

### 3.8. Senescence

Cd toxicity accelerates the ageing process in plant cells referred to as senescence. The Cd-induced morphological changes associated with senescence were shown in pine roots [16]. Cd exposure accelerated the senescence process of *Arabidopsis*, as indicated by an increase in SAG12 expression, a typical senescence marker gene [119].

Although senescence is a negative physiological process, it is one of the most important stages that plants undergo to maintain organ homeostasis and to escape from unfavourable conditions. Under biotic stress, SA is well known to induce senescence of infected tissues to build up a physical barrier against the spread of pathogens. In SA-deficient *Arabidopsis* plants, the transcript of SAG12 was considerably reduced or undetectable [88]. Whether treatment with Cd and SA manifests antagonistic or synergistic effects on plant senescence is still unclear. Generally, Cd distributes unevenly in field environment. Therefore, some parts of plant root may suffer Cd toxicity severely but some parts may not. An interesting hypothesis is that SA might accelerate Cd-induced senescence of stressed root parts, and then benefit the whole plant to elude Cd damage by adjusting the root growth direction towards the non-Cd contaminated environment. To test this hypothesis, a split-root experiment was conducted in which half of the roots were exposed to Cd while the other half were not exposed [67]. They found that Cd treatment caused senescence of the stressed part of the roots and stimulated the growth of the non-stressed part of the roots. However, SA pretreatment had no effect on the senescence of the Cd-exposed roots and did not lower the Cd uptake, which might be because the low level of exogenous SA used was insufficient to trigger senescence. Nonetheless, this hypothesis can be tested by using *Arabidopsis* mutants, such as *snc−1* with constitutively high concentrations of endogenous SA, since this mutant manifests a senescence phenotype and lower uptake of Cd compared to the wild-type plants [74].

## 4. Future Insights and Conclusions

Research over the past 20 years has strongly indicated that SA is a very promising molecule for the reduction of Cd toxicity in plants. Here, we reviewed reports describing the promoting role of SA in Cd resistance under various treatment methods, including pre-soaking, hydroponic exposure, and spraying. Figure 2 proposed the possible roles of SA in alleviating Cd toxicity to plants. However, there still remains a contradiction between the effects of SA at low and high doses. Furthermore, several unsolved and central questions concerning homeostasis, gene expression, and crosstalk with other phytohormones are still not fully understood.

### 4.1. SA Homeostasis

It is well known that plants can increase their intracellular concentration of SA to combat various environmental stresses, including Cd toxicity [120]. Exposure of Cd for 7 days increased the SA levels of leaves in young maize seedlings [121]. Compared with SA-deficient mutants, Cd stress significantly increased the SA level in wild-type *Arabidopsis*, indicating that Cd-induced SA synthesis occurs through the NahG- or SID2- pathways [72,74,75]. Exogenous SA treatment highlights the expression of *OsWRKY45* and increases the endogenous concentrations of SA in Cd-stressed rice and Lemna minor plants [57,66]. Under biotic stress, the mechanism of SA transport in plant cells has been illuminated. During pathogen attack, the hydroxyl group of SA is conjugated by glucose, resulting in formation of the SA glucoside (SAG). Then, SAG is actively transported from the cytosol into the vacuole, where it functions as an inactive storage form that can release free SA [122]. However, the balance of SA homeostasis in vivo under Cd stress still needs to be further monitored. 

### 4.2. SA-Related Gene Expression

The Cd toxicity and SA signal may share the same origins in regard to gene expression. Characterization and identification of the SA receptor during Cd stress are highly anticipated.

#### 4.2.1. Nonexpressor of Pathogenesis-Related (NPR) Protein

Nonexpressor of PR (NPR1 and NPR3/4), the canonical signal transducer of SA, regulates many gene expressions in seed germination, flowering, and senescence processes [123]. It has been uncovered that both NPR1 and NPR3/4 are SA receptors in regulating SA-mediated plant immunity [124]. Further genetic analysis indicated that NPR1 is a transcriptional activator while NPR3/4 are repressors functioning independent of NPR1 [125]. However, few studies have focused on the issue of whether SA-induced NPR is involved in Cd tolerance in plants. Surprisingly, it has been found that an increase in plant biomass was coupled with SA accumulation in *npr1−1* (a SA-deficient mutant) after 12-h-Cd exposure, suggesting that the regulation of Cd tolerance is not related to the NPR1 signalling pathway [74].

#### 4.2.2. Mitogen-Activated Protein Kinase (MAPK)

Another important example of SA-regulated gene expression involves mitogen-activated protein kinase cascades (MAPK), which acts as a negative regulator in plant growth and development in response to endogenous and environmental cues [126]. SA accumulation in *mpk4* mutants result in a severely dwarf phenotype [127]. Cd exposure quickly enhanced the kinase activity of MPK6, while the *mpk6* mutation enhanced Cd tolerance by alleviating oxidative stress [128]. However, until now, no studies have been conducted to evaluate the role of SA in signalling Cd resistance through MAPK regulation.

#### 4.2.3. ATP-Binding Cassette (ABC) Transporters

ATP-binding cassette (ABC) transporters belong to a large family that utilizes the energy of ATP binding and hydrolysis to transport elements across cellular membranes. In particular, they are involved in sequestering Cd or PC-Cd complexes into vacuoles to alleviate Cd toxicity [129]. An ABC transporter from soybean was identified that was strongly and rapidly induced by SA treatment [130]. Therefore, it is of interest to explore whether ABC transporters are involved in SA signalling and SA-induced Cd tolerance.

### 4.3. Crosstalk with Other Phytohormones

In addition to crosstalk with ROS transduction, the coordination between signalling of SA and other phytohormones is also an important aspect to consider [131]. SA pretreatment mitigated the Cd-induced disturbance in the levels of indoleacetic acid, cytokinins and ABA in wheat seedlings [98], and alleviated Cd toxicity in barley root tips by inhibiting auxin-mediated ROS generation [61]. It was also reported that SA combined with NO counteracted the negative effects of Cd on ryegrass plants [62]. However, the direct or indirect influence of SA signalling on the balance of plant hormones needs to be determined. 

In conclusion, investigations of SA signalling roles shed new light on the approaches to enhancing Cd tolerance with phytohormones. In addition to classical methods of exogenous application, these studies can now be complemented by the creation of a new generation of SA-excessive or SA-deficient mutants. Furthermore, with the characteristics of low cost and high efficiency, SA application shows promising use in helping plants defend against Cd toxicity. Seed presoaking and spraying with SA are pragmatic approaches for this purpose. In the meantime, novel roles of SA in Cd toxicity will likely continue to be unveiled.

## Figures and Tables

**Figure 1 ijms-20-02960-f001:**
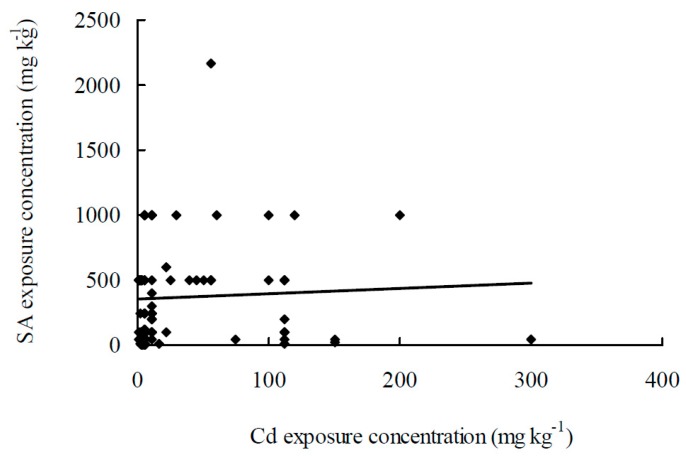
Representation of SA exposure concentrations plotted against Cd exposure concentrations, drawn from the data reported in Table 1. * Table 1 and Figure 1 are adapted from the Reference [76].

**Figure 2 ijms-20-02960-f002:**
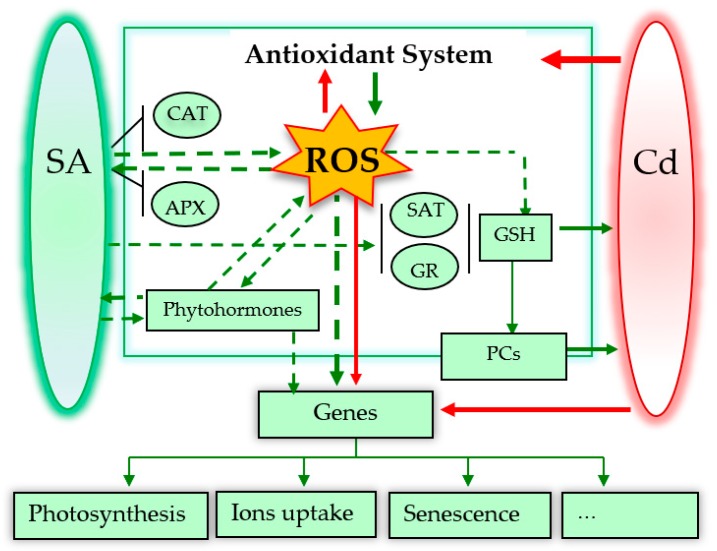
Possible roles of SA in alleviating Cd toxicity to plants. The dotted arrows mean possible signalling pathways. Red and green arrows indicate damage and positive effects, respectively. * Figure 2 is adapted from the reference [76].

**Table 1 ijms-20-02960-t001:** Effect of salicylic acid (SA) on cadmium (Cd) tolerance in plants.

	SA Treatment	Cd Treatment	Timeline	Plant Species	Main Responses * Means Negative or No Effect	Reference
Spraying	600 μM, 10 days	22.5 mg L^−1^	Simultaneous	Potato (*S. tuberosum* L.)	I, II, III, VII	[31]
	100 μM, 1 time	30, 60 and 120 mg kg^−1^ (pot)	Simultaneous	Peppermint (Mentha piperita)	I, II, III	[32]
	50 μM, 4 times in a 3-day interval	75, 150, and 300 mg kg^−1^	Simultaneous	Oilseed rape (*Brassica napus*)	II, III, V, VI	[33]
	10, 50, 100, and 200 μM each day treated for 50 mL last 4 days	44.8 mg kg^−1^	Pretreatment	Melon (*Cucumis melo* L.)	I, II, III	[34]
	500 μM, 1 time	40 mg kg^−1^	Pretreatment	Soybean (*Glycine max L. cv. Liaoxing 1*)	I, II, III	[35]
	2170 μM 1 time	56 and 112 mg kg^−1^	Simultaneous	Radish (*Raphanus sativus*)	I, *IV	[36]
	1000 μM for 10 mL, 45 times in a day interval	100 and 200 mg L^−1^	Simultaneous	Indian mustard (*Brassica juncea*)	I, II, III, IV, V	[37]
Presoaking	500 μM for 24 h.	112 mg L^−1^ for 72 h.	Pretreatment	Mungbean (*Vigna radiata* L. Wilczek)	I, II	[38]
	250 or 1000 μM for 8 h	5.6 and 11.2 mg L^−1^ for 10 days	Pretreatment	Flax (*Linum usitatissimum* L.)	Lipids	[39]
	250 or 1000 μM for 8 h	5.6 and 11.2 mg L^−1^ for 10 days	Pretreatment	Flax (*Linum usitatissimum* L.)	I, II	[40]
	500 μM for 12 h	0.56, 1.12, and 5.60 mg L^−1^ for 7 days	Pretreatment	Kentucky bluegrass	I, II, III, *IV, V	[41]
	500 μM for 12 h	56 and 112 mg kg^−1^ for 56 days	Pretreatment	Wheat (*Triticum aestivum* L. cv. Giza 168)	I, II, III, *IV	[42]
	500 μM for 6 h	1.12, 1.68, and 2.80 mg L^−1^ for 14 days	Pretreatment	Maize (*Zea mays*)	VI	[43]
	250 or 1000 μM for 8 h	5.6 and 11.2 mg L^−1^ for 10 days	Pretreatment	Flax (cv. Viking)	I, II	[44]
	250 or 1000 μM for 8 h	5.6 and 11.2 mg L^−1^ for 10 days	Pretreatment	Flax (*Linum usitatissimum* L.)	I, *IV, VI	[45]
	100 μM for 12 h	5.6 and 11.2 mg L^−1^ for 6 days	Pretreatment	Legume (*Phaseolus aureus* and *Vicia sativa*)	I, II	[46]
	250 and 500 μM for 12 h	5.6 mg L^−1^ for 12 days	Pretreatment	Bean (*R. communis* cv. Zibi 5)	I, III, IV	[30]
	250 and 1000 μM for 12 h	5.6 and 11.2 mg L^−1^ for 12 days	Pretreatment	Flax (*Linum usitatissimum* L.)	I, II, III, IV, V	[47]
	500 μM for 20 h	11.2, 44.8 and 112 mg kg^−1^ for 30 days	Pretreatment	Wheat (*Triticum aestivum* L.)	I, II, III	[48]
	100 μM for 3 h	3, 5, and 7 mg kg^−1^ for 3 days	Pretreatment	Soybean (Balkan, L608)	II, III, *IV	[49]
	500 μM for 6 h	25, 50, and 100 mg kg^−1^	Pretreatment	Hemp (*Cannabis sativa* L.)	I, II, III, *IV	[50]
	500 μM for 6 h	1.12, 1.68, and 2.80 mg L^−1^ for 14 days	Pretreatment	Maize (*Zea mays* L., hybrid Norma)	I, II, III, IV	[51]
	100 μM for 16 h	11.2 and 112 mg L^−1^ for 1 day	Pretreatment	Rice (cv: Longai)	I, II	[52]
	100 μM for 1, 3, 6 h	3 and 5 mg L^−1^ for 7 days	Pretreatment	Alfalfa (*Medicago sativa* L. cv. Evropa)	I, IV, V	[49]
	100 μM for 8 h	1.12, 11.2, and 112 mg L^−1^ for 1 day	Pretreatment	Rice (cv: Longai)	I, II, *IV	[53]
	500 μM for 6 h	2.8 mg L^−1^ for 12 days	Pretreatment	Barley (*Hordeum vulgare* cv Gerbel)	I, II, III, IV, V, VI, VII	[54]
Hydroponic application	10 μM for 15 days	16.8 mg L^−1^ for 15 days	Simultaneous	Rice (*Oryza sativa* L. Galileo))	I, II, III	[55]
	20 μM for 1 day	150 mg L^−1^ for 9 days	Pretreatment	*Nymphaea tetragona* Georgi	II, III, *IV,V	[56]
	50 μM for 7 days	1.12 mg L^−1^ for 7 days	Simultaneously	*Lemna minor*	II, III, IV, V	[57]
	50 μM for 1 day	11.2 mg L^−1^ for 8 h	Pretreatment	Wheat (*Triticum aestivum* L.)	I, *IV, Hormones	[33]
	500 μM for 24 h	56 mg L^−1^ for 1 day	Pretreatment	Maize (*Zea mays* L., hybrid Norma)	II, III, *IV, VI	[58]
	100, 200, 300 and 400 μM for 14 days	11.2 mg L^−1^ for 14 days	Simultaneous	Ryegrass (*Lolium perenne* L.)	I, II, III, *IV, V	[50]
	50 μM for 10 days	5.6 mg L^−1^ for 10 days	Simultaneous	Rice (*Oryza sativa* cv. HUR3022)	I, II, III	[59]
	100 μM for 14 days	22.4 mg L^−1^ for 14 days	Simultaneously	Peanut (Arachis hypogaea L.)	I, II, III, *IV, V	[60]
	250 and 500 μM for 10 mins	1.68 mg L^−1^ for 3 and 6 h	Post-treatment	Barley (*Hordeum vulgare* L.) cv. Slaven	I, II, Auxin	[61]
	200 μM for 14 days	11.2 mg L^−1^ for 14 days	Simultaneously	Ryegrass (*Lolium perenne* L.)	I, II, III, VI	[62]
	10, 50 and 100 μM for 7 days	2.24 mg L^−1^ for 3 days	Pretreatment	Bean (*Phaseolus vulgaris*)	I, II, III, IV, V	[63]
	60, 120, 250 and 500 mM	5.6 mg L^−1^ for 5 days	Pretreatment	Soybean (*Glycine max* L., A6445RG)	II, III, *IV, V, VI, VII (HO−1)	[64]
	1, 10, and 100 μM for 72 h	5.6 mg L^−1^ for 1 day	Pretreatment	Alfalfa (*Medicago sativa* L. cv Zhongmu No.1)	I, II, *IV, VI, VII (HO−1)	[65]
	3000 μM for 3 h	560 mg L^−1^ for 1 day	Pretreatment	Rice (*Oryza sativa L.,* cv. Taichung Native 1)	II, IV	[66]
	10 μM for 72 h	5.6 mg L^−1^ for 6 days	Pretreatment	Rice (*O. sativa* cv Jiahua 1)	I, II, IV	[67]
	10 μM for 72 h	5.6 mg L^−1^ for 6 days	Pretreatment	Rice (*O. sativa* cv Jiahua 1)	I, II	[68]
	10 μM for 24 h	5.6 mg L^−1^ for 6 days	Pretreatment	Rice (*O. sativa* cv Jiahua 1)	I, II, IV	[65]
	1, 10, and 100 μM	3 and 6 mg L^−1^ for 3 days	Simultaneous	Soybean (*Glycinemax* L. cv SG1)	*I, *IV, IV	[54]
	500 μM for 24 h	2.8 mg L^−1^ for 10 days	Pretreatment	Barley leaves (*Hordeum vulgare* cv Gerbel)	I, II, III, IV, V, VI, VII	[69]
	500 μM for 24 h	56 mg L^−1^ for 1 day	Pretreatment and simultaneously	Maize (*Zea mays* L., hybrid Norma)	*I, *II, *III, *VI	[70]
SA mutants	Up and down-regulating endogenesis SA	5.6 mg L^−1^ for 7 days	-	*NahG,snc1*	I, II, III, IV, VII	[71]
	Down-regulating endogenesis SA	0.56 mg L^−1^ for 12 days	-	*Sid2*	I, II, III, IV, V, VI, VII	[72]
	SA accumulation	16.8 mg L^−1^ for 28 days	-	*Lycium chinense*	II, III, IV, VII(*LcGSHS*)	[73]
	Up and down-regulating endogenesis SA	5.6, 11.2, and 16.8 mg L^−1^ for 7days	-	Accumulating mutant *snc1*, *npr1−1*, Reducing mutant *nahG*, *snc1/nahG*	*I, *II, *III	[74]
	Down-regulating endogenesis SA	56 mg L^−1^ for 5 days	-	*NahG*	*II, *III, *VII (CAT1)	[75]

I Growth, II antioxidant system, III photosynthesis, IV Cd uptake, V Ion uptake, VI phytochelatins, VII SA or Cd-induced genes.

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
