# Peer review of "Salicylic Acid Signals Plant Defence against Cadmium Toxicity"

_ijms, 2019, doi:10.3390/ijms20122960_

Round 1
Reviewer 1 Report
The review is about the crosstalk between cadmium and salicylic acid.
It is a nice review, well written.
I have only minoir remarks, for detail points.
Figure 1 should appear after table I.
For table I, the significance of I, II, III ... VII should be in the legends, at the top of the table.
Line 153, also quote sasek et al., 2014,abput a mutant that overaccumulates SA and that is dwarf. Quote also Janda and Ruelland, 2015, EEB.
Line 187 is a copy of the abstract of teh quoted paper. In the context of the abstact, the "double mutant analysis" may be clear. But here in this review it is not clear. Explanin: When pmr4 mutant was crossed wit mutants affected in SA sigballing pathway (NahG, pad4 mutants) it restored the susceptibility of pmr4 to fungus. something like that.
Author Response
Listed responses to the reviewers’ suggestions/comments on
IJMS-520440
Dear editor Dani Wu,
We appreciate the constructive comments from the reviewers and yourself. We would also like to take this opportunity to thank you for handling our manuscript (IJMS-520440).
According to the suggestion of the Journal, all revisions were clearly highlighted by using the "Track Changes" function in Microsoft Word. We now would like to list the point-to-point responses to the comments and/or suggestions raised by the reviewers as follows.
Reviewer #1
The review is about the crosstalk between cadmium and salicylic acid. It is a nice review, well written. I have only minor remarks, for detail points.
Point 1: Figure 1 should appear after table 1.
Response 1: According to the reviewer’s suggestion, we arranged the Figure 1 behind the table 1.
Point 2: For table I, the significance of I, II, III ... VII should be in the legends at the top of the table.
Response 2: According to the reviewer’s suggestion, we put the significance of I, II, III ... in the legends at the top of the table.
Point 3: Line 153, also quote sasek et al., 2014 about a mutant that overaccumulates SA and that is dwarf. Quote also Janda and Ruelland, 2015, EEB.
Response 3: Thanks for the reviewers’ constructive comments. We cited these two valuable references. Pleased to see Page 7, line 152 and line 155.
[47] Šašek,V.; Janda, M.; Delage, E.; Puyaubert, J.; Guivarc’h, A.; López Maseda, E.; Dobrev, P.I.; Caius, J.; Bóka, K.; Valentová, O.; et al. 2014. Constitutive salicylic acid accumulation in pi4kIIIβ1β2 Arabidopsis plants stunts rosette but not root growth. New Phytol. 2014, 203, 805–816
[48] Janda, M.; Ruelland, E. Magical mystery tour: salicylic acid signalling. Environ. Exp. Bot. 2015, 114, 117-128.
Point 4: Line 187 is a copy of the abstract of teh quoted paper. In the context of the abstract, the "double mutant analysis" may be clear. But here in this review it is not clear. Explain: When pmr4 mutant was crossed with mutants affected in SA signalling pathway (NahG, pad4 mutants) it restored the susceptibility of pmr4 to fungus. something like that.
Response 4: Thanks for the comments. We rewrited it. “A genetical research showed that pmr4 mutants are resistant to pathogens due to mutation of a callose synthase, while blocking SA signaling pathway restore the susceptibility, suggesting callose or callose synthase negatively regulates the SA pathway.” Pleased to see Page 8, line 188 to 190.
Reviewer 2 Report
The review "Salicylic acid signals plant defence against cadmium toxicity" by Guo et al. aims at giving an overview of the current knowledge of the impact of SA on Cd toxicity in plants. While the review is rather complete there is the air of plagiarism attached to it.
This year the first author published a book chapter that has a remarkable similarity with this manuscript. The table and the figures are (very-lightly) adapted version of what was published as "Role of Salicylic Acid in Mitigating Cadmium Toxicity in Plants" published in the book " Cadmium Toxicity and Tolerance in Plants". At least this should be made clear by adding in the legend of the figures and tables that they were adapated from this source, at least. How this should be done is up to the authors and it is up to the editor to judge whether this is a matter of misconduct or not, I can only signal the unreferenced similarity.
The authors do not mention the most important observation when looking at all the studies represented in table 1. Very few of these are actually relevant for coming to a solution for Cd-toxicity. It is interesting to see that hydroponic application of SA is better,
but Cd is never a problem in hydropony unless it is wanted. This is not a negative comment, this type of non-realistic studies is needed but the authors should mention it and discuss the data with this in mind.
Long-term studies on cadmium give a different and much less uniform image on the impact of Cd on plant physiology, increased lignification and pectin-sequestration. Yes it happens but not as clear as observed in short-term studies, for instance discussed in Parrotta et al, Frontiers in Plant Science 2015.
Line 117: it is not clear why this little data indicates that the presoaking treatment should be applied more carefully than the other methods. for all application methods negative effects are described in literature, not in all cases highlighted by the authors. Furthermore when reading line 296-298 presoaking is the only treatment that actually makes agricultural and/or economic sense, the others would only be interesting if there is a known, future cadmium contamination. Which never happens.
It is not clear why heading 3.7 is separataed from 3.6, furthermore focussing for instance cysteine metabolism with mentioning the change in expression of one of the enzymes in this pathway is not enough: acetyl serine is used in other pathways, cysteine can be synthesized in different ways, cysteine is an intermediate for methionine, .... Interesting, but unexplored by the authors is the impact of SA on S content (line 237-238) and its link with GSH (line 324-352).
Line 363-366: could the authors explain this hypothesis? From a biological point-of-view this makes little sense. Cd transport is from root-to-shoot, SA induced senescence would then cut off water uptake and lead to an accelerated dead of the plant.
Arabidopsis is in principle the scientific name, so it should be in italics in principle, what is more clear is that all scientific names must be in italics, also in the references.
The authors should be more correct in the wording, some examples but
others are found everywhere in the text, please do not just adjust the
few examples given below but do a detailed editing of the entire text
line 42: reactions must be replaced with mechanisms
line 47: catalyses must be replaced with converts
line 51-52, this is not a correct sentence
line 153: not a correct sentence
At several places in the text: superfluous means "unnecessary" replace with the correct term
line 183-184. not a correct sentence
line 186: "its important signaling role is alteration of cell wall components"?????
line
202: "exposure of SA to plants" must be "exposure of plants to SA",
furthermore the word exposure would be better replaced with treatment
Line 237: S composition, this should be S content
line 243: SA acts as an important signal in photosynthesis, not clear why, most SA effects are also observed in dark-grown plants, SA is a regulator of photosynthesis but probably indirect through its other effects. An idirect effect mentioned by the authors in the previous lines 241-242.
LIne 280: "SA signals the production of ROS" please clarify
Line 284-285: we can be sure that SA does not interfere with H2O2, SA may regulate H2O2 accumulation/detoxification
line 288: substrate dehydrogenises ????
line
319: if the studies are debatable they should not be mentioned, the
authors mean probably that the results fo these studies are
contradictory.
Line 426-427: "SA recovered Cd disturbance in auxins, cytokinins and ABA", please clarify
Author Response
Listed responses to the reviewers’ suggestions/comments on
IJMS-520440
Dear editor Dani Wu,
We appreciate the constructive comments from the reviewers and yourself. We would also like to take this opportunity to thank you for handling our manuscript (IJMS-520440).
According to the suggestion of the Journal, all revisions were clearly highlighted by using the "Track Changes" function in Microsoft Word. We now would like to list the point-to-point responses to the comments and/or suggestions raised by the reviewers as follows.
Reviewer #2
Point 1: The review "Salicylic acid signals plant defence against cadmium toxicity" by Guo et al. aims at giving an overview of the current knowledge of the impact of SA on Cd toxicity in plants. While the review is rather complete there is the air of plagiarism attached to it.
This year the first author published a book chapter that has a remarkable similarity with this manuscript. The table and the figures are (very-lightly) adapted version of what was published as "Role of Salicylic Acid in Mitigating Cadmium Toxicity in Plants" published in the book " Cadmium Toxicity and Tolerance in Plants". At least this should be made clear by adding in the legend of the figures and tables that they were adapted from this source, at least. How this should be done is up to the authors and it is up to the editor to judge whether this is a matter of misconduct or not, I can only signal the unreferenced similarity.
Response 1: According to the reviewer’s comments, we clarified that “Table 1 and Figure 1 are adapted from the following reference. Pleased to see Page 5 line 75.
[132] Guo B. Role of salicylic acid in mitigating cadmium toxicity in plants. In Cadmium Toxicity and Tolerance in Plants; Hasanuzzaman, M.; Prasad, M.N.V.; Fujita, M., Ed.; Elsevier: Amsterdam, Netherlands, 2019; Chapter 14, p 349-374.
It should be noted that there is a great difference between this review and the previous published “chapter”, and the “Similarity Index” is 2% checked by Turnitin (https://www.turnitin.com/).
(1) Difference in aims. This review contributes to the special issue “Salicylic Acid Signalling in Plants” of IJMS. Since the special issue mainly focuses on the mode of action of SA and rarely involves the phytotoxicology of Cd, this review detailed the Cd toxicity in plants, so that make it more readability on the modes of SA in Cd tolerance. In contrast, we do not extend the Cd toxicity in the “chapter” of the published book “Cadmium toxicity and tolerance in plants”, because the Cd toxicity has been illustrated very well.
(2) Difference in organization. Compared with the chapter, this review is an entirely new manuscript, which is arranged more methodically based on how Cd damage to plant from physiology to molecular biology.
(3) Difference in depth of review. According to the request of special issue, we made more efforts on the molecular mechanisms of SA-related signalling processes on Cd toxicity, and put more emphasis on recent findings especially for the mutant studies.
Point 2: The authors do not mention the most important observation when looking at all the studies represented in table 1. Very few of these are actually relevant for coming to a solution for Cd-toxicity. It is interesting to see that hydroponic application of SA is better, but Cd is never a problem in hydropony unless it is wanted. This is not a negative comment; this type of non-realistic studies is needed but the authors should mention it and discuss the data with this in mind.
Response 2: Thanks for the reviewer’s constructive comments. We clarified them accordingly.
Pleased to see Page 7, line 79 to 80. “However, very few of these literatures are actually relevant for coming to a solution for Cd toxicity”.
Page 7, line 117 to 119. “Indeed, hydroponic exposure of SA is less practicable because Cd is not a problem in the hydrographic environment. However, this type of studies is needed since SA or Cd treatment can be controlled more precisely and effectively, which is beneficial to the phytotoxicology research”
Point 3: Long-term studies on cadmium give a different and much less uniform image on the impact of Cd on plant physiology, increased lignification and pectin-sequestration. Yes it happens but not as clear as observed in short-term studies, for instance discussed in Parrotta et al, Frontiers in Plant Science 2015.
Response 3: Thanks. We added the reviewer’s constructive comments and recommended reference into the paper. “In addition, long-term Cd treatments may affect the synthesis and composition of cell wall, such as lignification, which in turn impact Cd sequestration more complicated [60].” Pleased to see Page 9, line 185 to 186.
[60] Parrotta, L.; Guerriero, G.; Sergeant, K.; Cai, G.; Hausman, J. F. Target or barrier? The cell wall of early- and later-diverging plants vs cadmium toxicity: differences in the response mechanisms. Front. Plant Sci. 2015, 6, 133.
Point 4: Line 117: it is not clear why this little data indicates that the presoaking treatment should be applied more carefully than the other methods. for all application methods negative effects are described in literature, not in all cases highlighted by the authors. Furthermore, when reading line 296-298 presoaking is the only treatment that actually makes agricultural and/or economic sense, the others would only be interesting if there is a known, future cadmium contamination. Which never happens.
Response 4: We agree with the reviewer, and deleted this unnecessary description. We also highlighted the practical use of presoaking method. Pleased to see Page 7, line 114 to 115. “It is worth mentioning that presoaking method is more practical for agricultural and/or economic use than the spray treatment.”
Point 5: (1)It is not clear why heading 3.7 is separated from 3.6, (2)furthermore focusing for instance cysteine metabolism with mentioning the change in expression of one of the enzymes in this pathway is not enough: acetyl serine is used in other pathways, cysteine can be synthesized in different ways, cysteine is an intermediate for methionine, .... (3)Interestingly, but unexplored by the authors is the impact of SA on S content (line 237-238) and its link with GSH (line 324-352).
Response 5: (1) For plants, one of the important detoxifying strategies is chelating Cd with GSH and PCs into the less sensitive organelles. SA may signal this process by regulating chelation metabolism, which is the main idea of heading 3.7. The discussion is started with “GSH” because GSH has dual roles as reducing equivalents and chelation, following on naturally from the 3.6. Thus, we think the purpose of 3.7 is different with 3.6, and emphasized this point in the heading title by “Glutathione and chelation”. Pleased to see Page 11, line 326
(2) Although many reports showed that simultaneous changes of the levels of SA and GSH, the mechanism of SA in regulating GSH synthesis under Cd stress is not fully understand. A few literatures have shown that SA may modulate the cysteine metabolism and SAT activity, which are the key processes for GSH synthesis. Indeed, this maybe not enough as the reviewer’s comments, but no other possible evidences are reported so far. Furthermore, we discussed the reports that SA may affect the activities of GR, GSHS, etc, which are also related GSH synthesis.
(3) The exogenous impact of SA on S content and its link with GSH was mentioned in Page 12 line 337 to 339. As the reviewer’s suggestion, we added the discussion in the paper. Pleased to see Page 12, line 343 to 346.
“Under Cd stress, although depletion of SA in sid2 mutants significantly enhanced the uptake of S, a key element for GSH construction, down-regulated transcription of SAT-c and SAT-p in sid2 blocked the process of GSH biosynthesis and resulted in lower GSH levels compared with the wild-type plants”
Point 6: Line 363-366: could the authors explain this hypothesis? From a biological point-of-view this makes little sense. Cd transport is from root-to-shoot, SA induced senescence would then cut off water uptake and lead to an accelerated dead of the plant.
Response 6: We are sorry for the unclear presentation. The real point is as follows: “Generally, Cd distributes unevenly in realistic soil environment. Therefore, some parts of plant root may suffer Cd toxicity but some parts may not. An interesting hypothesis is that SA might accelerated Cd-induced senescence of stressed root parts, and then benefit whole plant to elude Cd damage by adjusting the root growth direction towards the non-Cd contaminated environment.” Pleased to see Page 11, line 367 to 371.
Point 7: Arabidopsis is in principle the scientific name, so it should be in italics in principle, what is more clear is that all scientific names must be in italics, also in the references.
Response 7: According to the reviewer’s comments, all “Arabidopsis” in the paper were rechecked and corrected in italics, except for some references (19, 21, 35, 44, 45, 47, 80, 94, 96, 101, 106, and 110) which “Arabidopsis” are published in normal font.
Point 8: The authors should be more correct in the wording, some examples but others are found everywhere in the text, please do not just adjust the few examples given below but do a detailed editing of the entire text.
Response 8: According to the reviewer’s comments, we rechecked and corrected them throughout the whole paper, such as "exposure of plants to SA", “unnecessary”.
Point 9: line 42: reactions must be replaced with mechanisms
Response 9: We corrected it. Pleased to see Page 1, line 42.
Point 10: line 47: catalyses must be replaced with converts
Response 10: We corrected it. Pleased to see Page 2, line 47.
Point 11: line 51-52, this is not a correct sentence
Response 11: We rewrited it. “Another biosynthetic pathway is through isochorismate synthase that catalyses the conversion of chorismate into isochorismate.” Pleased to see Page 2, line 51 to 52.
Point 12: line 153: not a correct sentence
Response 12: We deleted the incorrect conclusion. “which might because SA activates the expression of senescence-related genes, such as SAG12 [46].” Pleased to see Page 7, line 152.
Point 13: At several places in the text: superfluous means "unnecessary" replace with the correct term
Response 13: We corrected it according to the reviewer’s comment. Pleased to see Page 7 line 165, Page 8 line 173, Page 9 line 261, and Page 11 line 320.
Point 14: line 183-184. not a correct sentence
Response 14: We rewrited it. “Findings between hyperaccumulating and non-hyperaccumulating ecotype of Sedm alfredii show that roots with higher cell-wall polysaccharides and activity of pectin methylesterase are more impermeable to Cd”. Pleased to see Page 8 Line 181 to 183.
Point 15: line 186: "its important signaling role is alteration of cell wall components"?????
Response 15: Sorry, we clarified it. “Whereas for SA, the signalling pathway is also involved in regulation of cell wall components.” Pleased to see Page 8 Line 187.
Point 16: line 202: "exposure of SA to plants" must be "exposure of plants to SA", furthermore the word exposure would be better replaced with treatment
Response 16: We corrected it according to the reviewer’s comment. Pleased to see Page 6 line 82, Page 7 line 123, and Page 8 line 204.
Point 17: Line 237: S composition, this should be S content
Response 17: We corrected it. Pleased to see Page 9 line 240.
Point 18: line 243: SA acts as an important signal in photosynthesis, not clear why, most SA effects are also observed in dark-grown plants, SA is a regulator of photosynthesis but probably indirect through its other effects. An indirect effect mentioned by the authors in the previous lines 241-242.
Response 18: We agree with the reviewer that SA is a regulator of photosynthesis but probably indirect through other effects. We cited a review reference [42] that has been discussed it detailly. Here, we concisely outlined the indirect effect of SA on photosynthesis, so that make more effort on the interaction of SA and Cd. Pleased to see Page 9 line 245-246. “SA may act as an important photosynthesis regulator by influence of RuBisCO activity, contribution to light acclimation and redox homeostasis, and function of stomatal switch.”
Point 19: Line 280: "SA signals the production of ROS" please clarify
Response 19: We clarified it. “It is well known that SA signals plant resistance through modulation of ROS metabolism.”
Pleased to see Page 10 line 282.
Point 20: Line 284-285: we can be sure that SA does not interfere with H2O2, SA may regulate H2O2 accumulation/detoxification.
Response 20: We corrected this sentence according to the reviewer’s comment. Pleased to see Page 10 line 287. “SA may regulate H2O2 accumulation through a self-amplifying feedback loop in Arabidopsis.”
Point 21: line 288: substrate dehydrogenises ????
Response 21: We are sorry for the spelling mistake, and corrected it in “substrate dehydrogenases” Pleased to see Page 10 line 291.
Point 22: line 319: if the studies are debatable they should not be mentioned, the authors mean probably that the results of these studies are contradictory.
Response 22: We corrected this sentence according to the reviewer’s comment. “The results of the SA mutants studies are still contradictory.” Pleased to see Page 12 line 321
Point 23: Line 426-427: "SA recovered Cd disturbance in auxins, cytokinins and ABA", please clarify
Response 23: We clarified it. “SA pretreatment mitigated the Cd-induced disturbance in the levels of indoleacetic acid, cytokinins and ABA in wheat seedlings” Pleased to see Page 13 line 431-432.

Round 2
Reviewer 2 Report
a minor check of the spelling would be good, but besides this the review is now acceptable for publication.